# Supplementation of Extender with Melatonin Improves the Motility, Mitochondrial Membrane Potential, and Fertilization Ability of Cryopreserved Brown-Marbled Grouper Sperm

**DOI:** 10.3390/ani14070995

**Published:** 2024-03-24

**Authors:** Qingxin Ruan, Sen Yang, Sijie Hua, Weiwei Zhang, Duo Li, Yang Yang, Xi Wang, Qinghua Wang, Zining Meng

**Affiliations:** 1State Key Laboratory of Biocontrol, Institute of Aquatic Economic Animals and Guangdong Province Key Laboratory of Aquatic Economic Animals, China-ASEAN Belt and Road Joint Laboratory on Mariculture Technology, School of Life Sciences, Sun Yat-Sen University, Guangzhou 510275, China; ruanqx3@mail2.sysu.edu.cn (Q.R.); huasj@mail2.sysu.edu.cn (S.H.); zhangww79@mail2.sysu.edu.cn (W.Z.); liduo7@mail2.sysu.edu.cn (D.L.); yangy888@mail.sysu.edu.cn (Y.Y.); wangx265@mail2.sysu.edu.cn (X.W.); wangqh55@mail2.sysu.edu.cn (Q.W.); 2College of Food Science and Technology, Guangdong Ocean University (Yangjiang Campus), Yangjiang 529599, China; yangsen@gdou.edu.cn; 3Southern Laboratory of Ocean Science and Engineering, Zhuhai 519000, China

**Keywords:** melatonin, antioxidants, sperm cryopreservation, *Epinephelus fuscoguttatus* (brown-marbled grouper), sperm quality

## Abstract

**Simple Summary:**

Sperm cryopreservation has been widely used to preserve the semen of high-quality males, and to a certain extent it has helped to solve the problem of grouper sperm supply. However, during this process, oxidative damage caused by reactive oxygen species (ROS) is considered to be one of the main causes of the decline in sperm quality. ROS are unstable and highly reactive, which can not only cause DNA damage and lipid peroxidation, but also lead to mitochondrial membrane depolarization and reduced ATP production, ultimately reducing sperm motility. Antioxidants can remove the ROS to protect the cells, a method that has been widely used in many species. In this study, we investigated if melatonin (MT) can be used as an effective antioxidant additive to reduce cryodamage to brown-marbled grouper (*Epinephelus fuscoguttatus*) and confirmed its optimal range. Our results showed that MT in the range of 0.1 to 0.25 mg/mL could improve the quality and fertilization ability of frozen–thawed brown-marbled grouper sperm. We anticipate that our results will have reference value for sperm cryopreservation and sample commercialization of groupers and other fish.

**Abstract:**

Sperm cryopreservation is a valuable tool for breeding, conservation, and genetic improvement in aquatic resources, while oxidative damage will cause a decline in sperm quality during this progress. Melatonin (MT), a natural antioxidant hormone, is used as an additive in sperm cryopreservation to reduce cellular damage from oxidative stress. Here, we aimed to investigate the effect of adding MT to the freezing medium in sperm cryopreservation of brown-marbled grouper (*Epinephelus fuscoguttatus*). Different concentrations of MT (0, 0.1, 0.25, and 0.5 mg/mL) were tested. We evaluated sperm motility, viability, apoptosis, mitochondrial membrane potential (MMP), and fertilization ability to assess the effects of MT supplementation. Our results demonstrated that the addition of MT to the extender improved the post-thaw motility, MMP, and fertilization ability of brown-marbled grouper sperm. The total motility, curvilinear velocity, straight linear velocity, and average path velocity in MT-treated groups (0.1 and 0.25 mg/mL) exhibited significantly higher values than that of the control group. A higher MMP (*p* < 0.05) was observed in the group treated with 0.25 mg/mL MT, suggesting that supplementation of MT in the extender might be able to protect mitochondrial membrane integrity effectively. Regarding fertilizing ability, 0.25 mg/mL MT yielded a significantly higher hatching rate than the control. An adverse effect was found with the concentration of MT up to 0.5 mg/mL, suggesting the possible toxicity of a high-dose addition. In this study, we optimized the sperm cryopreservation protocol of brown-marbled grouper, which might be valuable for sperm cryopreservation and sample commercialization of groupers and other fish.

## 1. Introduction

The brown-marbled grouper (*Epinephelus fuscoguttatus*) is an important reef fish mainly distributed in the Indo-Pacific region. Since the 1970s, it has become a common target of commercial export fisheries for the international live reef fish food trade, which caused a severe population decline. Brown-marbled grouper has been listed as vulnerable on the IUCN Red List of Threatened Species [1]. To reduce fishing pressure and meet international demand for reef fish food, this species has been extensively cultured in China and Southeast Asia. However, brown-marbled grouper is a typical protogynous hermaphroditic fish with a long reproductive cycle, resulting in a lack of mature male broodstock in hatchery practice [2]. Therefore, the insufficient supply of male reproductive gametes has become a major factor limiting the growth of brown-marbled grouper in aquaculture production.

Sperm cryopreservation can preserve the semen of high-quality males, solving the problem of sperm supply to a certain extent. Researchers have successfully developed sperm cryopreservation protocol for various kinds of grouper [2,3,4], which are used to deal with the asynchrony of sexual maturity of broodstocks, helping the crossbreeding of groupers. Researchers have obtained several hybridization varieties using cryopreserved sperm, such as *E. fuscoguttatus*♀ × *E. lanceolatus*♂ [5] and *E. lanceolatus*♂ × *E. akaara*♀ [6]. The most successful one among them, brown-marbled grouper♀ × giant grouper♂, has accounted for more than 70% of grouper market in China (over 19,000 tons in 2020), showing a promising promotion for the development of grouper industry [7,8,9,10]. However, during the process of sperm cryopreservation, oxidative damage caused by reactive oxygen species (ROS) is considered to be one of the main causes of the decline in sperm quality [11]. ROS are unstable and highly reactive; their production and accumulation in mitochondria can lead to mitochondrial membrane depolarization and reduced ATP production, ultimately reducing sperm motility [12]. To remove excessive ROS, seminal plasma contains various natural endogenous antioxidants and oxidant defensive enzymes, including ascorbic acid, carnitine, peroxidase, glutathione reductase superoxide dismutase, and so on [13]. However, the dilution of semen during sperm cryopreservation and the reduction in cytoplasm during the final stage of spermatogenesis can lead to a decrease in intracellular antioxidants, leading to insufficient antioxidant defense, and preventing sperm from responding quickly and effectively to ROS during cryopreservation [11]. Therefore, it is urgent to improve the antioxidant defense in the process of sperm freezing, and adding antioxidants to the cryopreservation medium is an effective method. Various enzymatic and non-enzymatic antioxidants have been tested as additives to improve sperm quality during cryopreservation [13,14]. Many of them have been proved to be effective in fish sperm cryopreservation. For instance, fetal bovine serum can significantly increase the motility of depik (*Rasbora tawarensis*) post-thaw sperm and protect the integrity of its DNA [15]; butylated hydroxytoluene (BHT) in doses of 0.001 mM can improve the progressive motility, duration of progressive motility, and fertilization rate of common carp (*Cyprinus carpio*) frozen–thawed sperm [16]. The reduced antioxidant properties of sperm during cryopreservation have been addressed by supplementing the freezing medium with antioxidants.

However, it is noteworthy that the effects of antioxidants may be species specific. For example, vitamin C significantly improved the frozen–thawed sperm motility, membrane and acrosome integrity in sturgeon (Yangtze sturgeon *Acipenser dabryanus*, Chinese sturgeon *A. sinensis* and Siberian sturgeon *A. baerii*) [14], but no inference was found in trout (Brook trout *Salvelinus fontinalis*, Arctic salmon *S. alpinus*, and rainbow trout *Oncorhynchus mykiss*) [17,18,19]. Furthermore, it seems to be dose-dependent for the antioxidant to protect the sperm from oxidative damage. Previous studies have shown that glutathione at 10 mg/L increased the motility of frozen–thawed sperm of cod, whereas at 50 mg/L the motility was not significantly different from that of the control [20]. These highlight the importance of finding a proper antioxidant supplementation and the proper concentration of it for practical production.

MT is a strong antioxidant that can not only directly remove ROS but also stimulate the rate-limiting enzyme γ-glutamyl cysteine synthetase and increasing intracellular GSH concentration, thus further reducing ROS levels [21]. MT was considered one of the most effective antioxidants to protect cells from oxidative stress, and has been successfully used as an additive for sperm cryopreservation in domestic animals and humans. MT has been found to be able to reduce the ROS level and the lipid peroxidation in post-thawed rabbit sperm [22]. Additionally, some of the metabolites produced when MT eliminates free radicals are thought to be antioxidative [23]. MT also plays a role in maintaining plasma membrane stability, preventing lipid membrane oxidation, and reducing transmembrane leakage [24]. Previous studies have shown that 1 mM (0.23 mg/mL) MT significantly increased plasma membrane integrity and decreased the rate of DNA damage in post-thawed rat sperm [25]. In the sperm cryopreservation of the Thai swamp buffalo bull (*Bubalus bubalis*), it has been proved that 1 mM (0.23 mg/mL) MT could improve the motility and viability of the post-thaw sperm [26]. Similarly, adding MT and myo-inositol in the extender could improve the post-cryopreservation quality of goat sperm, including motility, viability, and plasma membrane [27]. In rooster, post-thawed sperm with 1 mM (0.23 mg/mL) and 1μM (2.3 × 10^−4^ mg/mL) MT had higher total motility [28]. Additionally, it has been proved that 0.70 mg/mL MT could increase the TM and PM of human sperm and decrease the intracellular ROS levels [29]. Furthermore, MT has proven to be a useful application in the cryopreservation of the sperm of many mammals including human [30], sheep [31], horse [32], rabbit [33] and so on, and can help in removing ROS, stabilizing the plasma membrane, preventing the DNA damage and improving the membrane integrity. However, in comparison to mammals and humans, research on the effects of MT on fish sperm cryopreservation is relatively scarce [34,35].

Based on the protocol we conducted recently [3,36,37], this study was performed to investigate if MT can be used as an effective antioxidant additive to reduce cryodamage. In this study, we evaluated the motility, viability, apoptosis, and mitochondrial membrane potential (MMP) of frozen–thawed sperm, with different concentrations (0, 0.1, 0.25, and 0.5 mg/mL) of MT added to the freezing extender to prove that the addition of MT can increase the quality of frozen–thawed sperm. The hatching rate was also evaluated to assess the fertilization ability of sperm.

## 2. Materials and Methods

### 2.1. Broodstock and Gamete Collection

The experiments were conducted during the reproductive season (Jun to Aug) of brown-marbled grouper. Mature broodstocks (males: 9~10 years old, 7 ± 1 kg; females: 6 years old, 5 ± 1 kg) (Figure 1) were procured from a commercial hatchery operated by Hainan Chenhai Aquatic Co., Ltd. (108.64° E, 18.85° N) in Hainan Island, China. The fish were cultured in indoor recirculatory seawater system with water temperature maintained at 26 to 28 °C and exposed to natural photoperiods (14 h light and 10 h dark) (Figure 1). They were selected from pools and temporarily cultured in 2 pools (one for male and the other one for female) preparing for the experiment. They were fed with frozen fish every two days.

Before gamete collection, the fish were anesthetized with 0.1% tricaine methane sulfonate (MS-222, Sigma, St. Louis, MI, USA). The gametes were collected by gentle abdominal massage. Extreme care was taken to prevent seawater, urine, and feces contamination. Semen samples were collected in 50 mL tubes and stored at 4 °C before analysis. Eggs were collected in a clean plastic basin at room temperature. Artificial fertilization was performed within one hour after the egg collection.

### 2.2. Preparation and Cryopreservation of Samples

According to our previous study, a dilution ratio of 1:3 was employed with 0.3 M glucose as the extender and 10% DMSO (final concentration, *v*/*v*) as the cryoprotectant [3]. Semen samples from 6 fish were collected every time. Semen samples exhibiting total motility exceeding 90% were chosen to be cryopreserved. Gametes from at least 3 fish were mixed totally for every experiment to eliminate the differences caused by different fish.

The melatonin (solid powder, 98%, CAS 73-31-4, Macklin, Shanghai, China) was added to the extender to make the final concentration of 0 (control), 0.1, 0.25, and 0.5 mg/mL [38]. After mixing the extender and the cryoprotectant, the semen was added and gently mixed. The mixture was introduced into 100 μL straws (IMV Technologies, L’Agile, France) and cooled in a polyethylene foam box (internal dimensions 36 × 28 × 25 cm) with liquid nitrogen at a depth of 5 cm (Figure 1). The straws were frozen 3 cm above the surface of liquid nitrogen and stored in liquid nitrogen after cooling to below −80 °C (10 min). For thawing, the straws were thawed in a water bath at 40 °C for 7 s. Every group ended up with over 20 straws for the follow-up experiment. At least 3 straws were thawed for every single experiment.

### 2.3. Sperm Quality Evaluation

#### 2.3.1. Motility Parameters

Sperm motility parameters were evaluated using an Integrated Semen Analysis System (ISAS 2.0, Spain). After activation with fresh seawater, spermatozoa were loaded into a 10 μm depth chamber for motility analysis. Images of at least 3 fields and 500 spermatozoa (video sampling rate 25 frames per second) were collected for each sample. Total motility (TM, %), progressive motility (PM, %), curvilinear velocity (VCL, μm/s), straight linear velocity (VSL, μm/s), and average path velocity (VAP, μm/s) were recorded. Sperm with VCL > 10 μm/s was classified as motile [3]. Sperm with VAP > 50 μm/s and straightness > 80% were defined as progressive mobile [4]. In order to count the sperm concentration manually, semen was diluted 1000 times with the Ringer solution and subsequently placed in the Neubauer hemacytometer plate [4].

#### 2.3.2. Viability

Sperm viability was assessed based on plasma membrane integrity by double fluorescent staining using propidium iodide (PI) and SYBR-14 (L7011, Thermo Fisher Scientific, Waltham, MA, USA). Sperm suspension containing 10^6^ cells/mL in 100 μL was mixed with 0.5 μL of SYBR-14 (25 μM) and 0.5 μL PI (2.4 mM) adequately. After 10 min of incubation at 4 °C in the dark, 400 μL of phosphate buffered saline (PBS) was added for subsequent flow cytometry analysis. Spermatozoa with intact plasma membrane (SYBR-14^+^/PI^−^) were considered viable.

#### 2.3.3. Sperm Apoptosis Assay

Annexin V-FITC apoptosis detection kit (Beyotime, Shanghai, China) was used to identify apoptotic changes in sperm plasma membrane. The samples were washed twice with PBS and resuspended in 195 of μL binding buffer, stained with 5 μL of annexin V-FITC (about 25 μL for every million cells) and 10 μL of PI (about 50 μL for every million cells). After incubation at room temperature for 20 min in the dark, apoptotic cells (Annexin V-FIT C^+^/PI^−^) were detected by flow cytometry.

#### 2.3.4. Mitochondrial Membrane Potential (MMP)

Sperm mitochondrial activity was measured with the JC-1 Assay Kit (Beyotime, Shanghai, China). Semen was incubated at 37 °C for 20 min in the dark with 500 μL of JC-1 staining solution added (2.5 μL for every million cells). The cell suspension was subsequently washed twice with PBS and resuspended in 500 μL of assay buffer for further analysis. Sperm with red fluorescence (JC-1 aggregates) were considered to have high MMP.

#### 2.3.5. Flow Cytometry

Viability, apoptosis assay, and MMP were evaluated using a CytoFlex flow cytometer (Beckman Coulter, Indianapolis, IN, USA). For each assay, at least 10,000 sperm-specific events were recorded with flow rates ranging from 300 to 500 cells/s. Fluorescence probes were excited with a 488 nm laser (50 mW). Red fluorescence (PI, JC-1 aggregates) and green fluorescence (SYBR-14, JC-1 monomer, FITC) were captured through 585/42 nm filter and 525/40 nm filter, respectively. The data were analyzed using CytExpert (Beckman Coulter, Brea, CA, USA).

### 2.4. Artificial Fertilization Experiments

Eggs from three females were pooled and divided into petri dishes, about 400 eggs per dish (3 replicates per treatment, 12 dishes in total). Then, sperm was dropped on the eggs at a ratio of 10,000 spermatozoa per egg. After thorough mixture, 5 mL of fresh seawater was added into each dish to activate the gametes. After 10 min of sufficient exposure at room temperature, the eggs were then transferred into 500 mL round-bottom beakers. Half of the water in the beaker was exchanged every 8 h. The hatching rate was evaluated by counting the hatched larvae under a stereoscopic microscope after incubation for 24 h.

### 2.5. Statistical Analysis

All results were presented as means ± standard deviations and were analyzed using SPSS 21.0 (SPSS Inc., Chicago, IL, USA). The percentage data such as TM were transformed by arcsine square root. We used Shapiro–Wilk test to check whether the data fit the normal distribution. In order to explore the effect of melatonin supplementation on the quality and fertilization ability of post-thaw sperm of brown-marbled grouper, one-way ANOVA and Duncan’s multiple range tests were used. If the data did not conform to normality, the Kruskal–Wallis test was used for non-parametric testing. Pearson correlation test was used to check the correlation between the parameters. All of the significance levels were set at 0.05.

## 3. Results

### 3.1. Motility Parameters

The effects of different concentrations of melatonin on the frozen–thawed sperm motility parameters are shown in Table 1. Other than the fresh sperm, the highest TM and VCL were found in the 0.25 mg/mL group, reaching 81.16 ± 4.08% and 92.26 ± 2.13 μm/s, which were significantly different from those of the control group. In addition, the VCL of the 0.1 mg/mL group was also significantly higher than that of the control. The group with 0.1 mg/mL MT obtained the highest VSL and VAP (58.55 ± 3.32 μm/s and 70.63 ± 2.86 μm/s, respectively). The 0.5 mg/mL group had no significant difference with the control group in all parameters except VCL. The results of the Shapiro–Wilk test showed that one group of PM was not in line with normal distribution (*p* = 0.049 < 0.05), but the non-parametric test showed that there were differences in the distribution of PM among the groups (*p* = 0.06 > 0.05). Furthermore, a positive connection was found between TM and VCL (*p* < 0.01), TM and VSL (*p* < 0.05), PM and VAP (*p* < 0.05), VCL and VSL (*p* < 0.01), and VCL and VAP (*p* < 0.05).

### 3.2. Viability, Apoptosis Rate, and MMP

The effects of different melatonin concentration on the viability, apoptosis rate, and MMP of frozen–thawed brown-marbled grouper sperm are shown in Table 2. Melatonin in 0.25 mg/mL significantly increased the proportion of sperm cells with high MMP, which was 91.47 ± 3.91%. The results of flow cytometry were analyzed, as shown in Figure 2. It was found to have negative connection between apoptosis rate and VCL (*p* < 0.05).

### 3.3. Artificial Fertilization Experiments

The melatonin supplementation improved the hatching rate of thawed sperm (Figure 3), and the embryos and juveniles obtained from artificial fertilization with frozen–thawed sperm are shown in Figure 4. The highest hatching rate (86.3 ± 4.52%) was also reached in the 0.25 mg/mL group, while the other groups did not show significant differences. Additionally, it was found that hatching rates are positively connected with VCL (*p* < 0.01) and MMP (*p* < 0.05).

## 4. Discussion

During the process of sperm cryopreservation, the accumulation of ROS can cause various oxidative damage, affecting the quality and activity of sperm. Some studies have shown that higher ROS concentration can be detected in dead and morphologically abnormal sperm [39]. Cryopreservation of fish sperm with appropriate antioxidants can prevent oxidative damage [40]. MT has been proved to be an effective antioxidant in post-thaw sperm of various animals. For example, 1 μM (2.3 × 10^−4^ mg/mL) melatonin can improve the total motility, viability, and hatching rate of giant grouper (*Epinephelus lanceolatus*) frozen–thawed sperm [41]. It can also improve the motility, membrane integrity, acrosome integrity, fertilization rate, and birth rate of frozen–thawed sperm in rabbits [33]. However, there are few studies using MT as an antioxidant in the semen cryopreservation of different fish species. In this study, we verified that melatonin could improve the post-thaw sperm motility, MMP, and fertilization ability of brown-marbled grouper.

Sperm motility was considered to be the best biomarker of sperm quality, which has been found to be highly correlated with fertilization rate and hatching rate in some fish [42]. In this study, MT supplementation (0.25 mg/mL) significantly increased TM, while the addition of 0.1 mg/mL MT significantly increased VCL and VSL. Consistently with this, a previous study of *Prochilodus lineatus* showed that the addition of 2 mM (0.46 mg/mL) MT could significantly improve the VCL and VAP of frozen–thawed sperm, but did not change PM and VSL [43]. This result may be due to the ROS-scavenging effect of melatonin. In sperm cells, ATP was synthesized mainly in mitochondria, where ROS mainly accumulated [12]. As a water–oil amphiphilic hormone, MT can cross all of the morphological and physiological barriers in the cell and enter the mitochondria directly [34]. It has been found that adding MT would reduce the ROS in the post-thawed rabbit sperm [22]. At the same time, some of its metabolites, produced during this antioxidative process, were also thought to have antioxidant properties [23]. These properties allow melatonin to reduce mitochondrial damage caused by ROS accumulation and improve sperm motility [23,44].

Mitochondria are the energy supply organelles of the cell [45]. Most of the sublethal damage caused by cryopreservation is due to the alteration of mitochondrial function caused by the activation of the mitochondrial permeability transition pore (mPTP) [46]. The formation of mPTP can reduce MMP (ΔΨm) and promote the release of apoptotic factors, such as cytochrome C and pro-apoptotic factors, which can lead to cell apoptosis [47]. In this study, 0.25 mg/mL MT significantly increased the proportion of the sperm with high MMP, indicating that MT could play a role in reducing mPTP. It can improve MMP and protect sperm mitochondrial membrane. In our study on giant grouper, MT in 1 μM (2.3 × 10^−4^ mg/mL) and 2 μM (4.6 × 10^−4^ mg/mL) could also improve the proportion of the sperm with high MMP [41]. In previous studies, MT has proven to be able to improve the function of mitochondria in frozen–thawed sperm. It was confirmed that MT could inhibit the PPlase activity of Cyp D, which binds to ANT by activating the MT receptor 1 (MT 1) on ram sperm. It can inhibit the opening of mPTP in the mitochondrial membrane (PI2K- AKT-GSK 3β pathway) and the release of pro-apoptotic factors, improving the integrity of plasma membrane and MMP [31]. Additionally, the main function of mitochondria is to synthesize ATP via oxidative phosphorylation (OXPHOS) using the electron transport chain (ETC). MT was reported to be able to increase the mRNA expressions of mitochondrial respiratory complexes and the activities of complexes I, II, III, and IV in the frozen–thawed ram sperm, which play an important role in the ETC [45]. This reminds us that MT might be able to improve the OXPHOS of the frozen–thawed sperm, which may be caused by the inhibition of mPTP opening [48]. Additionally, it has been proved that MT can inhibit the intrinsic apoptotic pathway by inhibiting transport of CYT C into the cytoplasm. Using the AMPK/mTOR system, MT can effectively prevent the activation of the caspase apoptosis pathway, thereby reducing apoptosis in frozen–thawed sperm [49]. This suggests that melatonin might be able to reduce the rate of sperm apoptosis and improve viability. But, we did not find any significant difference in the viability and apoptosis rate, which is consistent with the results in *Brycon orbignyanus*: the viability was increased using 2 mM (0.46 mg/mL) MT, but no significant difference was found in the 1 mM (0.23 mg/mL) MT group [38]. The reason is thought to be that the concentration is not high enough to work, which is different from ours. In this study, it may be caused by the species-specific effect.

Fertilization and producing progeny are the utmost goals for sperm. As a result, hatching rate is widely used to evaluate the fertilization ability of sperm. In this study, 0.25 mg/mL MT significantly increased the hatching rate of frozen–thawed sperm of brown-marbled grouper, indicating that appropriate concentrations of MT addition can effectively improve the fertilization ability of frozen sperm in brown-marbled grouper. This may be related to the improvement of sperm motility and MMP as mentioned above. Likewise, it has been found that MT could improve the fertilization ability rate of frozen–thawed sperm of *Brycon orbignyanus*, which was improved by 1.06% to 8.40% after adding 2 mM (0.46 mg/mL) MT [50]. In addition to protecting mitochondria and inhibiting apoptosis, MT can also improve fertilization ability by reducing DNA fragmentation and lipid peroxidation in the frozen–thawed sperm [28,51]. Take rooster for example, MT in 1 mM, 1 μM, and 1 nM (0.23 mg/mL, 2.3 × 10^−4^ mg/mL and 2.3 × 10^−7^ mg/mL) decreased the DNA fragmentation (DFI) significantly [28]. It is also reported that 2 mM (0.46 mg/mL) MT can extend the motility time of the post-thawed sperm of *Brycon orbignyanus* [38], which leads to a higher possibility for sperm to reach the egg in the vitro fertilization, whether it is artificial or natural.

It is noteworthy that the protective effect of antioxidants against cryodamage may be shown in a species-specific manner. In our study, when 0.5 mg/mL MT was added in cryopreserved sperm, insignificant improvement was observed in all of the parameters we evaluated except VCL, which is consistent with the reports of the dose effect of other species. For instance, in a study about ram semen, 1 mM (0.23 mg/mL) MT showed the best results with significantly higher total motility, viability rates, progressive motility, intracellular ATP concentrations, and DNA integrity [51]. The best concentration for brown-marbled grouper sperm cryopreservation is different from that for the ram sperm cryopreservation. The beneficial effects (including motility, membrane and acrosome integrity) that vitamin C had on the frozen–thawed sperm of Yangtze sturgeon (*A. dabryanus*) [14] cannot be found in rainbow trout (*O. mykiss*) [17]. However, a high concentration melatonin was noneffective or even harmful to the sperm. In the research mentioned earlier studying ram sperm, no significant increase can be found in any of the indexes in the group of 10 mM (2.3 mg/mL), and the DNA linearity index decreased significantly [51]. In the short-term storage of rabbit semen, 1.5 mM (0.345 mg/mL) MT decreased the total motility, progressive motility, and viability of the sperm, but increased the rate of abnormalities, membrane damage, and acrosome damage [33]. To sum up, it is important to determine the best concentration of antioxidants for sperm cryopreservation. Consequently, our results suggest that the addition of an appropriate MT dose can improve the motility, MMP, and fertilization ability of brown-marbled grouper sperm, but a high concentration of MT might be toxic and increase the deformity rate and plasma membrane damage rate. Our study shows that the determination of an appropriate concentration of MT facilitates the sperm cryopreservation of brown-marbled grouper, which preserves the sperm of this threatened species for our germplasm bank.

## 5. Conclusions

The results of this study showed that adding melatonin to the sperm extender could improve the motility, mitochondrial membrane potential (MMP), and fertilization ability of cryopreserved brown-marbled grouper sperm after thawing. According to our results, it is recommended to add 0.1 mg/mL to 0.25 mg/mL melatonin for sperm cryopreservation of brown-marbled grouper. In order to prevent possible toxicity, it is suggested the addition of MT would be no more than 0.5 mg/mL. This study confirms that melatonin can be used as an effective antioxidant additive to the extender in the cryopreservation of brown-marbled grouper sperm, which might have reference value for sperm cryopreservation and sample commercialization of groupers and other fish.

## Figures and Tables

**Figure 1 animals-14-00995-f001:**
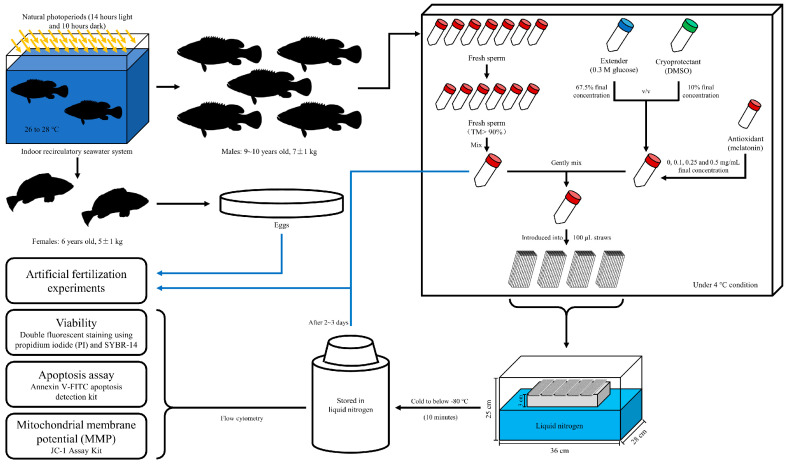
Broodstock, gamete collection, and experimental design.

**Figure 2 animals-14-00995-f002:**
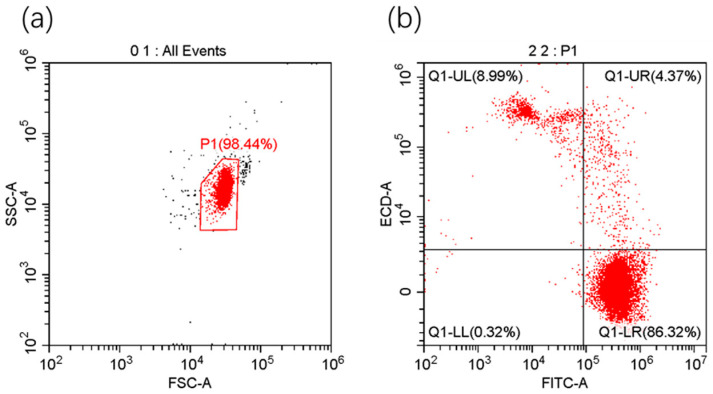
To evaluate viability, apoptosis rate, and MMP, flow cytometry experiments were taken. Take viability here as an example: (**a**) SSC-A and FSC-A were used to screen cell particles and exclude fragments. (**b**) Q1-UL: SYBR-14-/PI+; Q1-UR: SYBR-14+/PI+; Q1-LL: SYBR-14-/PI−; Q1-LR: SYBR-14+/PI−; spermatozoa in Q1-LR (SYBR-14+/PI−) were considered viable. Every experiment was repeated three times.

**Figure 3 animals-14-00995-f003:**
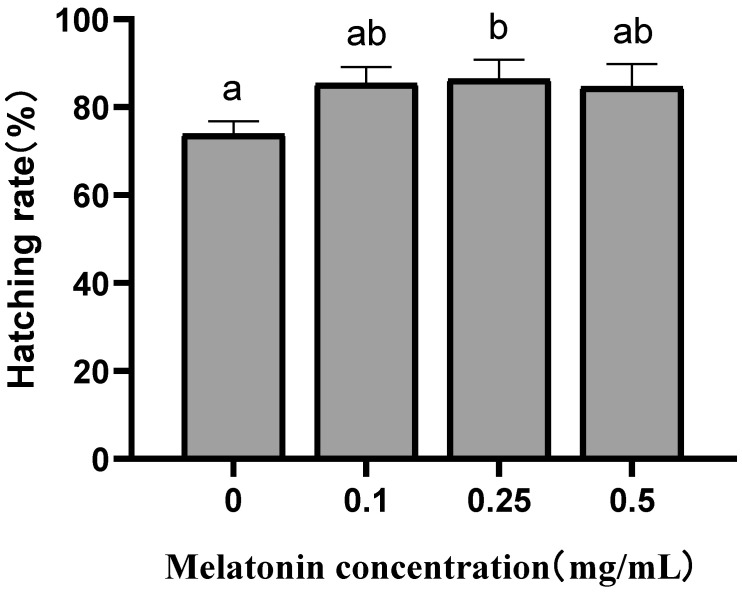
Effects of different concentrations of melatonin on the hatching rate of frozen–thawed sperm from brown-marbled grouper. All results were presented as means ± standard deviations. Different superscripts mean significant differences under the condition of *p* < 0.05.

**Figure 4 animals-14-00995-f004:**
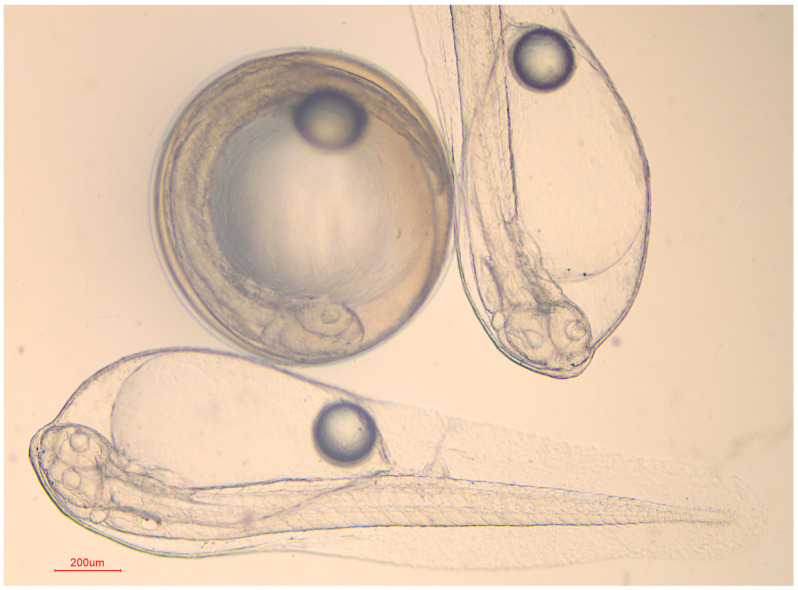
Embryos and larvae from artificial fertilization with frozen–thawed sperm of brown-marbled grouper.

**Table 1 animals-14-00995-t001:** The motility parameters of frozen–thawed sperm with different concentrations of melatonin and fresh sperm of brown-marbled grouper.

Melatonin (mg/mL)	TM (%)	PM (%)	VCL (μm/s)	VSL (μm/s)	VAP (μm/s)
Fresh	93.85 ± 0.94 ^a^	28.12 ± 3.10	124.62 ± 6.01 ^a^	44.29 ± 2.91 ^a^	67.66 ± 4.25 ^ab^
0 (Control)	76.50 ± 1.10 ^c^	23.88 ± 2.35	85.81 ± 5.01 ^c^	49.23 ± 6.89 ^ac^	62.56 ± 6.54 ^b^
0.1	78.86 ± 3.59 ^bc^	28.28 ± 3.77	91.90 ± 2.65 ^b^	58.55 ± 3.32 ^b^	70.63 ± 2.86 ^a^
0.25	81.16 ± 4.08 ^b^	29.49 ± 1.98	92.26 ± 2.13 ^b^	51.96 ± 6.02 ^c^	67.00 ± 4.73 ^ab^
0.5	76.49 ± 1.23 ^c^	26.56 ± 3.01	90.35 ± 3.35 ^ab^	50.39 ± 5.22 ^ac^	65.27 ± 4.49 ^ab^

TM, total motility; PM, progressive motility; VCL: curvilinear velocity; VSL: straight linear velocity; VAP: average path velocity. All results were presented as means ± standard deviations. Different superscripts in the same column indicated significant differences (*p* < 0.05). The results of Shapiro–Wilk test showed that one group of PM was not in line with normal distribution.

**Table 2 animals-14-00995-t002:** Effects of different concentrations of melatonin on viability, cell apoptosis, and MMP of frozen–thawed spermatozoa of brown-marbled grouper.

Melatonin (mg/mL)	Viability (%)	Apoptosis Rate (%)	High MMP (%)
0	85.8 ± 1.5	16.8 ± 2.1	77.1 ± 6.6 ^a^
0.1	86.4 ± 1.6	15.3 ± 0.6	89.0 ± 5.0 ^ab^
0.25	87.4 ± 1.3	15.7 ± 1.1	91.5 ± 3.9 ^b^
0.5	83.6 ± 1.4	15.9 ± 2.9	86.1 ± 4.8 ^ab^

MMP: mitochondrial membrane potential. All results were presented as means ± standard deviations. Different superscripts in the same column of the table indicated significant differences under the condition of *p* < 0.05.

## Data Availability

Data are contained within the article and Appendix A.

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
