# Peer review of "Supplementation of Extender with Melatonin Improves the Motility, Mitochondrial Membrane Potential, and Fertilization Ability of Cryopreserved Brown-Marbled Grouper Sperm"

_animals, 2024, doi:10.3390/ani14070995_

Round 1
Reviewer 1 Report (Previous Reviewer 2)
Comments and Suggestions for Authors
I thank the authors for making the requested modifications, which clearly improved the manuscript. My final recommendation for the authors is to modify the following more accurately:
Lines 177 - 181: describe further details of the kit used, especially the final concentrations.
The purpose of a detailed description of the protocols performed is that they can be repeated by other researchers.
Response: Thanks for your hard working and advice, we have added the final concentration of PI and SYBR-14(about 50 μL for every million cells) in Line 177, V, FITC (about 25 μL for every million cells) and PI (about 50 μL each for every million cells) in Line 185 and JC-1 (2.5 μL for every million cells) in Line 191 in the revised version.
Reviewer's response: When I mention the need to describe concentrations more accurately, I am referring to indicating at what final concentration the staining or reagent will be in the tube containing the cells, rather than specifying how many microliters, as this measure is subjective. Please see the following example:
“For laser confocal microscopy analysis, sperm were incubated with 0.01 mM calcein-AM and 0.4 mM cobalt chloride at 378C for 15 min, to obtain sperm with fluorescence only in the midpiece, and 1 mM propidium iodide was added to discard dead spermatozoa.”
Author Response
Thanks for your affirmation and suggestion. We have added the final concentration of SYBR-14 and PI in Line 178, but the exact concentrations of Annexin V-FITC and PI in the Annexin V-FITC apoptosis detection kit (Beyotime, Shanghai, China) and JC-1 in the JC-1 Assay Kit (Beyotime, Shanghai, China) are unavailable, which we can only describe in this way. We believe that other researchers will be able to repeat our experiments by using the same type of kits from the same supplier.
Thanks very much again for taking your time to review this manuscript and your affirmation. It would be our honor to publish this article in Animals.
Reviewer 2 Report (Previous Reviewer 1)
Comments and Suggestions for Authors
The authors addressed most of my previous comments somewhat satisfactorily. I have no have additional request or comments.
Author Response
Thanks very much for taking your time to review this manuscript and your affirmation. It would be our honor to publish this article in Animals.
This manuscript is a resubmission of an earlier submission. The following is a list of the peer review reports and author responses from that submission.
Round 1
Reviewer 1 Report
Comments and Suggestions for Authors
In your manuscript entitled "Supplementation of extender with melatonin improves the motility, mitochondrial membrane potential and fertility of cryopreserved brown-marbeled grouper sperm", you present a study on the effect of melatonin addition to extender prior to grouper sperm cryopreservation. You add different concentrations of melatonin to the extender and see minor but significant changes in sperm quality and hatch rate.
The study can be interesting to specific readers of "Animals" specifically interested in marine fish aquaculture procedures. Your study is relatively straight forward and the results are clearly presented, but the study design or its description has some flaws. For example, melatonin was added to the cryopreservation extender, but it is not described in which kind of solution or preparation the melatonin was originally present, where the melatonin originated from (company, stock number etc.) and which additives or other substances might have been present in the melatonin preparation. Also, a hatching test is perfomed to show the percent of hatched eggs after fertilization with sperm cryopreserved with different concentrations of melatonin, but it is not shown which percentage of the hatchlings survive later on, ideally to the time point at which they would usually be harvested. It makes no difference in grouper production if the hatch rate in the melatonin-treated group is a bit higher, but the amount/number of grouper harvested stays the same. As you specifically aim at improving the production rate in grouper aquaculture, this study falls somewhat short of really showing that melatonin in sperm cryopreservation makes a difference.
Please add more data on the melatonin and on the fish "production" rate after hatching for clarification.
Author Response
Manuscript Number: animals-2748256
Dear editor and reviewer:
Thanks very much for taking your time to review this manuscript. I really appreciate all your comments and suggestions! We have completely revised the manuscript according to the reviewers’ comments and the changes have been marked by revise tracks in the revised manuscript. The following is the response for reviewer’s comments. Please find my itemized responses in below.
Best Regards,
Dr. Zining Meng
To Reviewer #1:
In your manuscript entitled "Supplementation of extender with melatonin improves the motility, mitochondrial membrane potential and fertility of cryopreserved brown-marbled grouper sperm", you present a study on the effect of melatonin addition to extender prior to grouper sperm cryopreservation. You add different concentrations of melatonin to the extender and see minor but significant changes in sperm quality and hatch rate.
Response: Thank you for your decision and constructive comments on my manuscript. After adding different concentrations of melatonin to the extender, significant changes can be found in sperm quality and hatching rate in our study. We think that we have found an optimal range of melatonin supplement which is 0.1 to 0.25 mg/mL.
The study can be interesting to specific readers of "Animals" specifically interested in marine fish aquaculture procedures. Your study is relatively straight forward and the results are clearly presented, but the study design or its description has some flaws. For example, melatonin was added to the cryopreservation extender, but it is not described in which kind of solution or preparation the melatonin was originally present, where the melatonin originated from (company, stock number etc.) and which additives or other substances might have been present in the melatonin preparation. Also, a hatching test is performed to show the percent of hatched eggs after fertilization with sperm cryopreserved with different concentrations of melatonin, but it is not shown which percentage of the hatchlings survive later on, ideally to the time point at which they would usually be harvested. It makes no difference in grouper production if the hatch rate in the melatonin-treated group is a bit higher, but the amount/number of grouper harvested stays the same. As you specifically aim at improving the production rate in grouper aquaculture, this study falls somewhat short of really showing that melatonin in sperm cryopreservation makes a difference.
Please add more data on the melatonin and on the fish "production" rate after hatching for clarification.
Response: Thanks for your recognition and interest of our study.
Thank you very much for your kind reminder. These information (solid powder, 98%, CAS 73-31-4, Macklin, Shanghai, China) have added in Line 151.Considering its high purity, melatonin play a major role in antioxidant process.
Hatching rate is able to prove the quality of the sperm, which means the ability to fertilize fish eggs. Previous studies have shown that hatching rate is an important index and it is widely used to show the improvement of fertilization ability (Ding et al. 2009, Palhares et al. 2020, Yang et al. 2022). Therefore, we mainly focused on the hatching rate, which has been keenly indicated to production rate. Unfortunately, the whole process from incubation to cultivation need a lot of time. We will improve our experimental conditions and consider it in our future studies. Thank you very much for your helpful advice.
Reference:
Ding, S., J. Ge, C. Hao, M. Zhang, W. Yan, Z. Xu, J. Pan, S. Chen, Y. Tian, and Y. Huang. 2009. Long-term cryopreservation of sperm from Mandarin fish Siniperca chuatsi. Anim Reprod Sci 113:229-235.
Palhares, P. C., I. L. Assis, J. Souza, T. S. Franca, R. C. Egger, D. A. J. Paula, and L. D. S. Murgas. 2020. Effect of melatonin supplementation to a cytoprotective medium on post-thawed Brycon orbignyanus sperm quality preserved during different freezing times. Cryobiology 96:159-165.
Yang, S., B. Fan, X. Chen, Y. Hua, and Z. Meng. 2022. Optimization of a sperm cryopreservation protocol for giant grouper (Epinephelus lanceolatus). Aquaculture 555.
Reviewer 2 Report
Comments and Suggestions for Authors
Title
“Supplementation of extender with melatonin improves the motility, mitochondrial membrane potential and fertility of cryopreserved, brown-marbled sperm.”
The title is informative, it is not too long, but I recommend the authors to describe differently the following phrase "and fertility of cryopreserved …." perhaps it would be more appropriate to say, "fertilizing ability".
Abstract
Line 22: the authors point out that "the addition of the extender with MT could improve the....." if they already have results they should say "the addition of the extender with MT improve the......".
Line 17 - 19: although the general objective is understood, it is possible to improve the wording.
Introduction
Lines 113 - 116: The introduction is clear and supports the study well, but I recommend the authors to eliminate this part as it corresponds to the material and methods section.
Material and methods
In both the "introduction" and "discussion" sections the authors describe MT concentrations in mM, but in the "material and methods" section they are in mg/mL. I recommend the authors to change the unit of the data in the "results" section of the manuscript to make the discussion of the results clearer. According to my calculations it would be 0.43 mM - 2.1 mM.
Lines 159 - 161: how did the authors count motile spermatozoa, or does ringer's solution immobilize them?
Line 166: what was the final concentration of PI and SYBR-14 used by the authors?
Lines 173-174: what was the final concentration of annexin V, FITC and PI used by the authors?
Lines 177 - 181: describe further details of the kit used, especially the final concentrations.
The purpose of a detailed description of the protocols performed is that they can be repeated by other researchers.
Results
However, the results are clearly described, I recommend the authors to add a representative image of each experiment, mainly for flow cytometry.
Discussion
The main problem of the discussion is that the authors contrast results expressed in different units.
Linea 262: The authors state that “In consistent with this, a previous study in Prochilodus lineatus showed the addition of 2.0 mM MT could significantly improve the VCL and VAP….” But according to my calculations 2.0 mM is more like 0.5 mg/mL and not 0.25 mg/mL as described by the authors.
I encourage the authors to review this data and edit the discussion and conclusions accordingly.
Author Response
Manuscript Number: animals-2748256
Dear editor and reviewer:
Thanks very much for taking your time to review this manuscript. I really appreciate all your comments and suggestions! We have completely revised the manuscript according to the reviewers’ comments and the changes have been marked by revise tracks in the revised manuscript. The following is the response for reviewer’s comments. Please find my itemized responses in below.
Best Regards,
Dr. Zining Meng
To Reviewer #2:
Title
“Supplementation of extender with melatonin improves the motility, mitochondrial membrane potential and fertility of cryopreserved, brown-marbled sperm.”
The title is informative, it is not too long, but I recommend the authors to describe differently the following phrase "and fertility of cryopreserved …." perhaps it would be more appropriate to say, "fertilizing ability".
Response: Thanks very much for your time and concentration on our manuscript. We have reconsidered the title and exchanged the word “fertility” to “fertilization ability” in the revised manuscript.
Abstract
Line 22: the authors point out that "the addition of the extender with MT could improve the....." if they already have results they should say "the addition of the extender with MT improve the......".
Response: Thank you very much for your advice. We have changed “could improve” to “improved” in Line 26-27 in the revised version.
Line 17 - 19: although the general objective is understood, it is possible to improve the wording.
Response: Thanks, we have deleted the sentence in Line 18 as “Sperm cryopreservation is a valuable tool for breeding, conservation and genetic improvement in aquatic resources, while oxidative damage will cause the decline in sperm quality during this progress.”
Introduction
Lines 113 - 116: The introduction is clear and supports the study well, but I recommend the authors to eliminate this part as it corresponds to the material and methods section.
Response: Thank you for your decision and constructive comments on my manuscript. We have rewritten this paragraph to make it suitable for the introduction part in Lines 120-126 as followed:
“Based on the protocol we conducted recently, this study was conducted to investigate if MT can be used as an effective antioxidant additive to reduce cryodamage. In this study, we evaluated the motility, viability, apoptosis and mitochondrial mem-brane potential (MMP) of frozen-thawed sperm, with different concentrations (0, 0.1, 0.25 and 0.5 mg/mL) of MT added to the freezing extender to prove that the addition of MT can increase the quality of frozen-thawed sperm. The hatching rate was also evaluated to assess the fertilization ability of sperm.”
Material and methods
In both the "introduction" and "discussion" sections the authors describe MT concentrations in mM, but in the "material and methods" section they are in mg/mL. I recommend the authors to change the unit of the data in the "results" section of the manuscript to make the discussion of the results clearer. According to my calculations it would be 0.43 mM - 2.1 mM.
Response: Thanks for your advice. Our selections of concentration were made depend on the result of our preliminary experiment, which used the unit “mg/mL”, but the papers we mentioned in “introduction” and “discussion” used “mM”. We finally standardized the unit of concentration as mg/mL in the whole manuscript.
Lines 159 - 161: how did the authors count motile spermatozoa, or does ringer's solution immobilize them?
Response: Thank you. We use Integrated Semen Analysis System (ISAS 2.0, Spain) to count the motile sperm. Sperm with VCL (curvilinear velocity) >10 μm/s was classified as motile, which would be recognized and counted automatically by the ISAS system. This method is mentioned in Line 163-172. Fish sperm was immotile before activation, but sperm was motile when adding activation solution to activate it. For example, seawater is able to activate the sperm of seawater fish due to its high osmotic pressure. In this study, the brown-marbled grouper cannot be activated by ringer`s solution that is low osmotic pressure, which keep them immotile helping to count.
Line 166: what was the final concentration of PI and SYBR-14 used by the authors?
Lines 173-174: what was the final concentration of annexin V, FITC and PI used by the authors?
Lines 177 - 181: describe further details of the kit used, especially the final concentrations.
The purpose of a detailed description of the protocols performed is that they can be repeated by other researchers.
Response: Thanks for your hard working and advice, we have added the final concentration of PI and SYBR-14(about 50 μL for every million cells) in Line 177, V, FITC (about 25 μL for every million cells) and PI (about 50 μL each for every million cells) in Line 185 and JC-1 (2.5 μL for every million cells) in Line 191 in the revised version.
Results
However, the results are clearly described, I recommend the authors to add a representative image of each experiment, mainly for flow cytometry.
Response: Thanks for your advice. We have added an image of flow cytometry in Line 261 in the revised version to show our flow cytometry experiment.
Discussion
The main problem of the discussion is that the authors contrast results expressed in different units.
Response: Thanks for your advice, we have standardized the unit of concentration as mg/mL in the whole manuscript.
Linea 262: The authors state that “In consistent with this, a previous study in Prochilodus lineatus showed the addition of 2.0 mM MT could significantly improve the VCL and VAP….” But according to my calculations 2.0 mM is more like 0.5 mg/mL and not 0.25 mg/mL as described by the authors.
Response: Thank you for your advice. As mentioned in the introduction, the suitable concentration of the same antioxidant in different species are different. The main reason of mentioning this research is to prove that the melatonin can improve the motility parameters of frozen-thawed sperm.
I encourage the authors to review this data and edit the discussion and conclusions accordingly.
Response: Thanks for your advice, we have revised based on your comments, including improve the wording, standardizing the unit and complementing the details of the experiments.
Reviewer 3 Report
Comments and Suggestions for Authors
The authors evaluated the motility, viability, apoptosis and mitochondrial membrane potential of frozen-thawed sperm of brown-marbled grouper, with different concentrations (0, 0.1, 0.25 and 0.5 mg/mL) of melatonin added to the freezing extender.
The authors also assessed the hatching rate during the study, which increases the validity of the results.
Major issues
-Please provide details of the selection procedure for individuals included in the study. How did you perform randomization? Please describe the selection procedure and the selection criteria. Also, please include a table with inclusion and exclusion criteria. The information is paramount, but is not presented in the manuscript.
-Please carry out and present the results of analysis of correlation between the various parameters of semen evaluation studied in this work. Findings should be included in a new sub-section in Results.
Minor issues
-Introduction. Please include a brief summary of relevant studies performed in mammals.
-Discussion. Please include a new sub-section with clinical applications of the Results.
Overall
Revision after taking into consideration the above comments and reevaluation.
Author Response
Manuscript Number: animals-2748256
Dear editor and reviewer:
Thanks very much for taking your time to review this manuscript. I really appreciate all your comments and suggestions! We have completely revised the manuscript according to the reviewers’ comments and the changes have been marked by revise tracks in the revised manuscript. The following is the response for reviewer’s comments. Please find my itemized responses in below.
Best Regards,
Dr. Zining Meng
To Reviewer #3:
The authors evaluated the motility, viability, apoptosis and mitochondrial membrane potential of frozen-thawed sperm of brown-marbled grouper, with different concentrations (0, 0.1, 0.25 and 0.5 mg/mL) of melatonin added to the freezing extender.
The authors also assessed the hatching rate during the study, which increases the validity of the results.
Response: Thank you for your decision and constructive comments on my manuscript. It`s our pleasure to have your approval and advices.
Major issues
-Please provide details of the selection procedure for individuals included in the study. How did you perform randomization? Please describe the selection procedure and the selection criteria. Also, please include a table with inclusion and exclusion criteria. The information is paramount, but is not presented in the manuscript.
Response: Thank you for your advice. We have added this part in the revised paper in Line 149-150. We selected the sperm exhibiting total motility exceeding 90% (n=6) for cryopreservation and mixed the gametes from at least 3 fishes every time to eliminate the differences caused by different fish.
-Please carry out and present the results of analysis of correlation between the various parameters of semen evaluation studied in this work. Findings should be included in a new sub-section in Results.
Response:
Thanks for your valuable advice, we have added this part in Line 220-229 as followed:
We evaluated 5 motility parameters, viability, apoptosis rate, MMP and hatching rate in this research. TM is the percentage of moving sperm, while PM is the percentage of progressively moving sperm. These two and viability are used to show how many sperm is still alive and be able to move progressively. VCL is the curvilinear velocity of the sperm who are still moving, while VSL is the straight linear velocity and VAP is the average path velocity (average of VSL). These there are used to show how fast the alive sperm moves to check the quality of the sperm. Apoptosis rate is used to confirm if MT can decrease the cell apoptosis, while MMP is to show whether MT has a positive effect on mitochondrial membrane. The artificial fertilization experiments and hatching rate is to evaluate the fertilization ability of the frozen-thawed sperm.
Minor issues
-Introduction. Please include a brief summary of relevant studies performed in mammals.
Response: Thanks. We have added a summary in Line 111-117 as followed:
“Furthermore, MT has been proved to be a useful application in cryopreservation of sperm of many mammals including human (Deng et al. 2017), sheep (Fang et al. 2020), house(Lançoni et al. 2018), rabbit (Fadl et al. 2021) and so on, which can help removing ROS, stabilizing the plasma membrane, preventing the DNA damage and improving the membrane integrity.”
-Discussion. Please include a new sub-section with clinical applications of the Results.
Response: Thank you for your advice, we have rewritten the last paragraph (Line 359-384) of the discussion to describe the clinical applications as followed:
“It is noteworthy that the protective effect of antioxidant against cryodamage showed a species-specific manner. In our study, when 0.5 mg/mL MT was added in cryopreserved sperm, insignificant improvement was observed in all of the parameters we evaluated except VCL, which is consistent with the reports of the dose effect of other species. For instance, in a study about ram semen, 0.23 mg/mL showed the best results with significantly higher total motility, viability rates, progressive motility, intracellular ATP concentrations and DNA integrity (Succu et al. 2011). The best concentration for brown-marbled grouper sperm cryopreservation is different from that for the ram sperm cryopreservation. The beneficial effects (including motility, membrane and acrosome integrity) of vitamin C had on the frozen-thawed sperm of Yangtze sturgeon (Acipenser dabryanus) (Li et al. 2018) cannot be found in rainbow trout (Oncorhynchus mykiss) (Lahnsteiner et al. 2011). However, high concentration melatonin was noneffective or even harmful to the sperm. In the research mentioned earlier studying on ram sperm, no significant increase can be found in all of the indexes in the group of 2.3 mg/mL, and the DNA linearity index decreased significantly (Succu et al. 2011). In the short-term storage of rabbit semen, 0.345 mg/mL1.5 mM MT decreased the total motility, progressive motility and viability of the sperm, but increased the rate of abnormalities, membrane damage and acrosome damage (Fadl et al. 2021). To sum up, it is important to determine the best concentration of antioxidant for sperm cryopreservation. Consequently, our results suggest that the addition of appropriate MT can improve the motility, MMP and fertilization ability of brown-marbled grouper sperm, but high concentration of MT might be toxic and increase the deformity rate and plasma membrane damage rate. Our study that the determination of appropriate concentration of MT facilitates the sperm cryopreservation of brown-marbled grouper, which preserve the sperm of this threatened species for our germplasm bank.”
Overall
Revision after taking into consideration the above comments and reevaluation.
Response: Thank you for your advices, we have revised based on your comments.
Reference:
Deng, S.-L., T.-C. Sun, K. Yu, Z.-P. Wang, B.-L. Zhang, Y. Zhang, X.-X. Wang, Z.-X. Lian, and Y.-X. Liu. 2017. Melatonin reduces oxidative damage and upregulates heat shock protein 90 expression in cryopreserved human semen. Free Radical Biology and Medicine 113:347-354.
Fadl, A. M., A. R. M. Ghallab, M. M. Abou-Ahmed, and A. R. Moawad. 2021. Melatonin can improve viability and functional integrity of cooled and frozen/thawed rabbit spermatozoa. Reprod Domest Anim 56:103-111.
Fang, Y., C. Zhao, H. Xiang, G. Jia, and R. Zhong. 2020. Melatonin improves cryopreservation of ram sperm by inhibiting mitochondrial permeability transition pore opening. Reprod Domest Anim 55:1240-1249.
Lahnsteiner, F., N. Mansour, and F. A. Kunz. 2011. The effect of antioxidants on the quality of cryopreserved semen in two salmonid fish, the brook trout (Salvelinus fontinalis) and the rainbow trout (Oncorhynchus mykiss). Theriogenology 76:882-890.
Lançoni, R., E. C. C. Celeghini, M. B. R. Alves, K. M. Lemes, A. M. Gonella-Diaza, L. Z. Oliveira, and R. P. d. Arruda. 2018. Melatonin added to cryopreservation extenders improves the mitochondrial membrane potential of postthawed equine sperm. . Journal of Equine Veterinary Science 69:78-83.
Li, P., M. D. Xi, H. Du, X. M. Qiao, Z. G. Liu, and Q. W. Wei. 2018. Antioxidant supplementation, effect on post-thaw spermatozoan function in three sturgeon species. Reprod Domest Anim 53:287-295.
Succu, S., F. Berlinguer, V. Pasciu, V. Satta, G. G. Leoni, and S. Naitana. 2011. Melatonin protects ram spermatozoa from cryopreservation injuries in a dose-dependent manner. J Pineal Res 50:310-318.
Round 2
Reviewer 3 Report
Comments and Suggestions for Authors
The authors have addressed all the points raised and have made changes in the manuscript, which improve the quality.
The manuscript can be accepted.